# Robust Models are less Over-Confident

**Julia Grabinski**
Fraunhofer ITWM, Kaiserslautern
Visual Computing, University of Siegen
`julia.grabinski@itwm.fraunhofer.de`

**Paul Gavrikov**
IMLA, Offenburg University

**Janis Keuper**
Fraunhofer ITWM, Kaiserslautern
IMLA, Offenburg University

**Margret Keuper**
University of Siegen
Max Planck Institute for Informatics
Saarland Informatics Campus Saarbrücken

## Abstract

Despite the success of convolutional neural networks (CNNs) in many academic benchmarks for computer vision tasks, their application in the real-world is still facing fundamental challenges. One of these open problems is the inherent lack of robustness, unveiled by the striking effectiveness of adversarial attacks. Current attack methods are able to manipulate the network's prediction by adding specific but small amounts of noise to the input. In turn, adversarial training (AT) aims to achieve robustness against such attacks and ideally a better model generalization ability by including adversarial samples in the trainingset. However, an in-depth analysis of the resulting robust models beyond adversarial robustness is still pending. In this paper, we empirically analyze a variety of adversarially trained models that achieve high robust accuracies when facing state-of-the-art attacks and we show that AT has an interesting side-effect: it leads to models that are significantly less overconfident with their decisions, even on clean data than non-robust models. Further, our analysis of robust models shows that not only AT but also the model's building blocks (like activation functions and pooling) have a strong influence on the models' prediction confidences.
**Data & Project website:** `https://github.com/GeJulia/robustness_confidences_evaluation`

## 1  Introduction

Convolutional Neural Networks (CNNs) have been shown to successfully solve problems across various tasks and domains. However, distribution shifts in the input data can have a severe impact on the prediction performance. In real-world applications, these shifts may be caused by a multitude of reasons including corruption due to weather conditions, camera settings, noise, and maliciously crafted perturbations to the input data intended to fool the network (adversarial attacks). In recent years, a vast line of research (e.g. [25, 36, 44]) has been devoted to solving robustness issues, highlighting a multitude of causes for the limited generalization ability of networks and potential solutions to facilitate the training of better models.

A second, yet equally important issue that hampers the deployment of deep learning based models in practical applications is the lack of calibration concerning prediction confidences. In fact, most models are overly confident in their predictions, even if they are wrong [31, 45, 57]. Specifically, most conventionally trained models are unaware of their own lack of expertise, i.e. they are trained to make confident predictions in any scenario, even if the test data is sampled from a previously unseen domain. Adversarial examples seem to leverage this weakness, as they are known to not only fool the

36th Conference on Neural Information Processing Systems (NeurIPS 2022).

network but also to cause very confident wrong predictions [46]. In turn, adversarial training (AT) has shown to improve the prediction accuracy under adversarial attacks [22, 25, 65, 87]. However, only few works so far have been investigating the links between calibration and robustness [45, 60], leaving a systematic synopsis of adversarial robustness and prediction confidence still pending.

In this work, we provide an extensive empirical analysis of diverse adversarially robust models with regard to their prediction confidences. Therefore, we evaluate more than 70 adversarially robust models and their conventionally trained counterparts, which show low robustness when exposed to adversarial examples. By measuring their output distributions on benign and adversarial examples for correct and erroneous predictions, we show that adversarially trained models have benefits beyond adversarial robustness and are less over-confident.

To cope with the lack of calibration in conventionally trained models, Corbière et al. [13] propose to rather use the true class probability than the standard confidence obtained after the Softmax layer, such as to circumvent the overlapping confidence values for wrong and correct predictions. However, we observe that exactly these overlaps are an indicator for insufficiently calibrated models and can be mitigated by the improvement of CNNs building blocks, namely downsampling and activation functions, that have been proposed in the context of adversarial robustness [17, 28].

Our work analyzes the relationship between robust models and model confidences. Our experiments for 71 robust and non-robust model pairs on the datasets CIFAR10 [43], CIFAR100 and ImageNet [19] confirm that non-robust models are overconfident with their false predictions. This highlights the challenges for usage in real-world applications. In contrast, we show that robust models are generally less confident in their predictions, and, especially CNNs which include improved building blocks (downsampling and activation) turn out to be better calibrated manifesting low confidence in wrong predictions and high confidence in their correct predictions. Further, we can show that the prediction confidence of robust models can be used as an indicator for erroneous decisions. However, we also see that adversarially trained networks (robust models) overfit adversaries similar to the ones seen during training and show similar performance on unseen attacks as non-robust models. Our contributions can be summarized as follows:

- We provide an extensive analysis of the prediction confidence of 71 adversarially trained models (**robust models**), and their conventionally trained counterparts (**non-robust models**). We observe that most non-robust models are exceedingly over-confident while robust models exhibit less confidence and are better calibrated for slight domain shifts.

- We observe that specific layers, that are considered to improve model robustness, also impact the models' confidences. In detail, improved downsampling layers and activation functions can lead to an even better calibration of the learned model.

- We investigate the detection of erroneous decisions by using the prediction confidence. We observe that robust models are able to detect wrong predictions based on their confidences. However, when faced with unseen adversaries they exhibit a similarly weak performance as non-robust models.

Our analysis provides a first synopsis of adversarial robustness and model calibration and aims to foster research that addresses both challenges jointly rather than considering them as two separate research fields. To further promote this research, we released our modelzoo[1].

## 2 Related Work

In the following, we first briefly review the related work on model calibration which motivates our empirical analysis. Then, we revise the related work on adversarial attacks and model hardening.

**Confidence Calibration.** For many models that perform well with respect to standard benchmarks, it has been argued that the robust or regular model accuracy may be an insufficient metric [2, 13, 18, 79], in particular when real-world applications with potentially open-world scenarios are considered. In these settings, reliability must be established which can be quantified by the prediction confidence [58]. Ideally, a reliable model would provide high confidence predictions on correct classifications, and low confidence predictions on false ones [13, 57]. However, most networks are not able to

---

[1] https://github.com/GeJulia/robustness_confidences_evaluation

instantly provide a sufficient calibration. Hence, confidence calibration is a vivid field of research and proposed methods are based on additional loss functions [32, 35, 45, 48, 52], on adaptions of the training input by label smoothing [54, 60, 63, 75] or on data augmentation [20, 45, 76, 88]. Further, [58] present a benchmark on classification models regarding model accuracy and confidence under dataset shift. Various evaluation methods have been provided to distinguish between correct and incorrect predictions [13, 56]. Naeini et al. [56] defined the networks *expected calibration error* (ECE) for a model $f$ by with $0 \leq p \leq \infty$

$$\text{ECE}_p = \mathbb{E}[|\hat{z} - \mathbb{E}[1_{\hat{y}=y}|\hat{z}]|^p]^{\frac{1}{p}} \tag{1}$$

where the model $f$ predicts $\hat{y} = y$ with the confidence $\hat{z}$. This can be directly related to the over-confidence $o(f)$ and under-confidence $u(f)$ of a network as follows [81]:

$$|o(f)\mathbb{P}(\hat{y} \neq y) - u(f)\mathbb{P}(\hat{y} = y)| \leq \text{ECE}_p, \tag{2}$$

where [55]

$$o(f) = \mathbb{E}[\hat{z}|\hat{y} \neq y] \quad u(f) = \mathbb{E}[1 - \hat{z}|\hat{y} = y], \tag{3}$$

i.e. the over-confidence measures the expectation of $\hat{z}$ on wrong predictions, under-confidence measures the expectation of $1 - \hat{z}$ on correct predictions and ideally both should be zero. The ECE provides an upper bound for the difference between the probability of the prediction being wrong weighted by the networks over-confidence and the probability of the prediction being correctly weighted by the networks under-confidence and converges to this value for the parameter $p \to 0$ (in eq. 1). We also recur to this metric as an aggregate measure to evaluate model confidence. Yet, it should be noted that the ECE metric is based on the assumption that networks make correct as well as incorrect predictions. A model that always makes incorrect predictions and is less confident in its few correct decisions than it is in its many erroneous decisions can end up with a comparably low ECE. Therefore, ECE values for models with an accuracy below 50% are hard to interpret.

Most common CNNs are over-confident [31, 45, 57]. Moreover, the most dominantly used activation in modern CNNs [34, 39, 69, 73] remains the ReLU function, while is has been pointed out by Hein et al. [35] that ReLUs cause a general increase in the models' prediction confidences, regardless of the prediction validity. This is also the case for the vast majority of the adversarially trained models we consider, except for the model by [17] to which we devote particular attention.

**Adversarial Attacks.** Adversarial attacks intentionally add perturbations to the input samples, that are almost imperceptible to the human eye, yet lead to (high-confidence) false predictions of the attacked model [25, 53, 74]. These attacks can be classified into two categories: white-box and black-box attacks. In black-box attacks, the adversary has no knowledge of the model intrinsics [4], and can only query its output. These attacks are often developed on surrogate models [10, 42, 78] to reduce interaction with the attacked model in order to prevent threat detection. In general, though, these attacks are less powerful due to their limited access to the target networks. In contrast, in white-box attacks, the adversary has access to the full model, namely the architecture, weights, and gradient information [25, 44]. This enables the attacker to perform extremely powerful attacks customized to the model. One of the earliest approaches, the *Fast Gradient Sign Method* (FGSM) by [25] uses the sign of the prediction gradient to perturb input samples into the direction of the gradient, thereby increasing the loss and causing false predictions. This method was further adapted and improved by *Projected Gradient Descent* (PGD) [44], *DeepFool* (DF) [53], *Carlini and Wagner* (CW) [5] or *Decoupling Direction and Norm* (DDN) [65]. While FGSM is a single-step attack, meaning that the perturbation is computed in one single gradient ascent step limited by some $\epsilon$ bound, multi-step attacks such as PGD iteratively search perturbations within the $\epsilon$-bound to change the models' prediction. These attacks generally perform better but come at an increased cost of the attack. *AutoAttack* [14] is an ensemble of different attacks including an adaptive version of PGD, and has been proposed as a baseline for adversarial robustness. In particular, it is used in robustness benchmarks such as RobustBench [15].

**Adversarial Training and Robustness.** To improve robustness, adversarial training (AT) has proven to be quite successful on common robustness benchmarks. Some attacks can be simply defended by using their adversarial examples in the training set [25, 65] through an additional loss [22, 87]. Furthermore, the addition of more training data, by using external data, or data augmentation techniques such as the generation of synthetic data, has been shown to be promising for more robust models [6, 26, 27, 62, 68, 80]. RobustBench [15] provides a leaderboard to study the improvements made by the aforementioned approaches in a comparable manner in terms of their robust accuracy.

Madry et al. [50] observed that the performance of adversarial training depends on the models' capacity. High-capacity models are able to fit the (adversarial) training data better, leading to increased robust accuracy. Later research investigated the influence on increased model width and depth [26, 85], and quality of convolution filters [24]. Consequently, the best-performing entries on RobustBench [15] are often using Wide-ResNet-70-16's or even larger architectures. Besides this trend, concurrent works also started to additionally modify specific building blocks of CNNs [17, 29]. Grabinski et al. [28] showed that weaknesses in simple AT, like FGSM, can be overcome by improving the network's downsampling operation.

**Adversarial Training and Calibration.** Only a few but notable prior works such as [45, 60] have investigated adversarial training with respect to model calibration. Without providing a systematic overview, [45] show that AT can help to smoothen the prediction distributions of CNN models. Qin et al. [60] investigate adversarial data points generated using [5] with respect to non-robust models and find that easily attackable data points are badly calibrated while adversarial models have better calibration properties. In contrast, we analyze the robustness and calibration of pairs of robust and non-robust versions of the same models rather than investigating individual data points. [77] introduce an adversarial calibration loss to reduce the calibration error. Further, [72] propose confidence calibrated adversarial training to force adversarial samples to show uniform confidence, while clean samples should be one hot encoded. Complementary to [15], we provide an analysis of the predictive confidences of adversarially trained, robust models and release conventionally trained counterparts of the models from [15] to facilitate future research on the analysis of the impact of training schemes versus architectural choices. Importantly, our proposed large-scale study allows a differentiated view on the relationship between adversarial training and model calibration, as discussed in Section 3. In particular, we find that adversarially trained models are not always better calibrated than vanilla models especially on clean data, while they are consistently less over-confident.

**Adversarial Attack Detection.** A practical defense besides adversarial training, can also be established by the detection and rejection of malicious input. Most detection methods are based on input sample statistics [23, 30, 33, 37, 47, 49], while others attempt to detect adversarial samples via inference on surrogate models, yet these models themselves might be vulnerable to attacks [12, 51]. While all of these approaches perform additional operations on top of the models' prediction, we show that simply taking the models' prediction confidence can be used as a heuristic to reject erroneous samples.

## 3 Analysis

In the following, we first describe our experimental setting in which we then conduct an extensive analysis on the two CIFAR datasets with respect to robust and non-robust model[2] confidence on clean and perturbed samples as well as their ECE. Further, we observe by computing the ROC curves of these models that robust models are best suited to distinguish between correct and incorrect predictions based on their confidence. In addition we point out that the improvement of pooling operations or activation functions within the network can enhance the models' calibration further. Last, we also investigate ImageNet as a high resolution dataset and observe that the model with the highest capacity and AT can achieve the best performance results and calibration.

### 3.1 Experimental Setup

We have collected 71 checkpoints of robust models [1, 3, 7–9, 11, 16, 17, 21, 22, 26, 27, 38, 40, 41, 59, 61, 62, 64, 67, 68, 70, 71, 80, 83, 84, 86, 87, 89, 90] listed on the $\ell_\infty$-RobustBench leaderboard [15]. Additionally, we compare each appearing architecture to a second model *trained without AT or any specific robustness regularization, and without any external data* (even if the robust counterpart relied on it). Training details can be found in appendix A.

Then we collect the predictions alongside their respective confidences of robust and non-robust models on clean validation samples, as well as on samples attacked by a white-box attack (PGD), and a black-box attack (Squares). PGD (and its adaptive variant APGD [14]) is the most widely used white-box attack and adversarial training schemes explicitly (when using PGD samples for

---

[2]The classification into robust and non-robust models is based on the models' robustness against adversarial attacks. We consider a model to be robust when it achieves considerably high accuracy under AutoAttack [14].

training) or implicitly (when using the faster but strongly related FGSM attack samples for training) optimize for PGD robustness. In contrast, the *Squares* attack alters the data at random with an allowed budget until the label flips. Such samples are rather to be considered out-of-domain samples even for adversarially trained models and provide a proxy for a model's generalization ability. Thus, *Squares* can be seen as unseen attack for all models while PGD might be not for some adversarially trained, robust models.

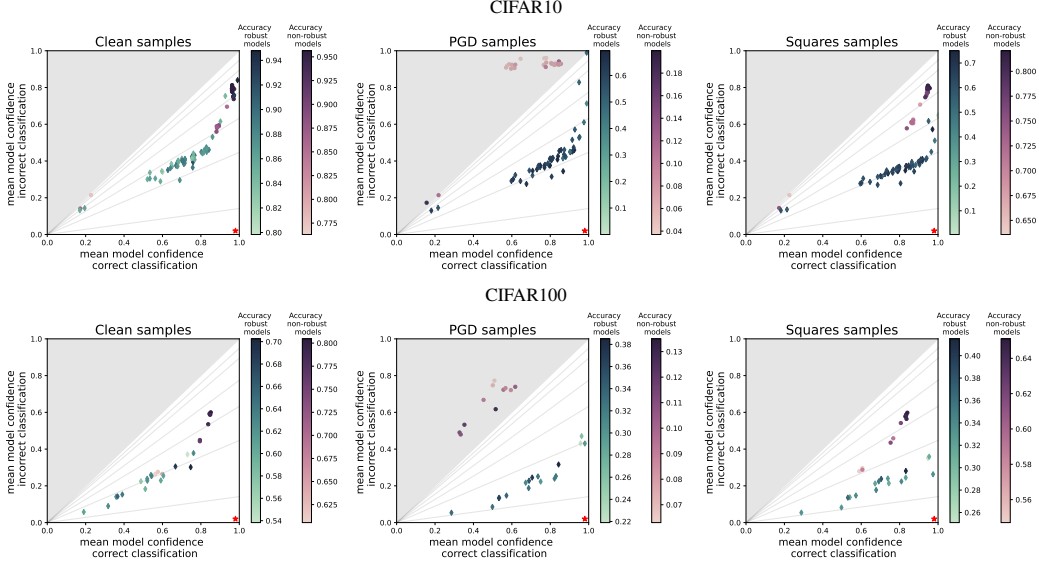

Figure 1: Mean model confidences on their correct (x-axis) and incorrect (y-axis) predictions over the full CIFAR10 dataset (top) and CIFAR100 dataset (bottom), clean (left) and perturbed with the attacks PGD (middle) and Squares (right). Each point represents a model. Circular points (purple color-map) represent non-robust models and diamond-shaped points (green color-map) represent robust models. The color of each point represents the models accuracy, darker signifies higher accuracy (better) on the given data samples. The star in the bottom right corner indicates the optimal model calibration and the gray area marks the area were the confidence distribution of the network is worse than random, i.e. more confident in incorrect predictions than in correct ones.

## 3.2 CIFAR Models

**CIFAR10** [43] is a simple ten class dataset consisting of 50,000 training and 10,000 validation images with a resolution of $32 \times 32$. Since it is significantly cheaper to train on CIFAR10 in comparison to e. g. ImageNet, and its low resolution allows to discount additional costs of adversarial training, most entries on RobustBench [15] focus on CIFAR10.

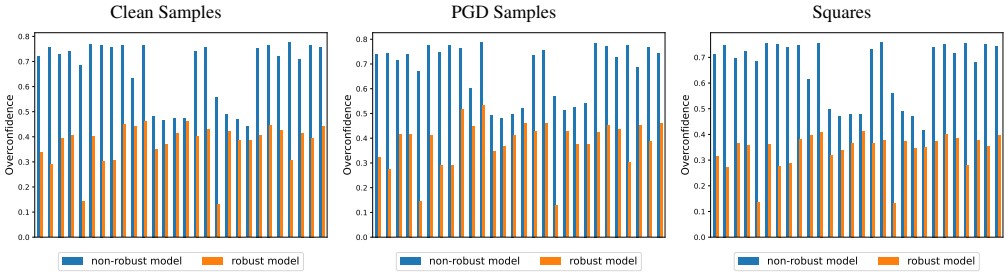

Figure 2: Overconfidence (lower is better) bar plots of robust models and their non-robust counterparts trained on CIFAR10. Non-robust models are highly overconfident, in contrast, their robust counterparts are less over-confident.

Figure 1 shows an overview of all robust and non-robust models trained on CIFAR10 in terms of their accuracy as well as their confidence in their correct and incorrect predictions. Along the isolines, the ratio between confidence in correct and incorrect predictions is constant. The gray

area indicates scenarios where models are even more confident in their incorrect predictions than in their correct predictions. Concentrating on the models' confidence, we can see that robust models (marked by a diamond) are in general less confident in their predictions, while non-robust models (marked by a circle) exhibit high confidence in all their predictions, both correct and incorrect. This indicates that non-robust models are not only more susceptible to (adversarial) distribution shifts but are also highly over-confident in their false predictions. Practically, such behaviour can lead to catastrophic consequences in safety-related, real-world applications. Robust models tend to have lower average confidence and a favorable confidence trade-off even on clean data (Figure 1, top left). When adversarial samples using PGD are considered (Figure 1, top middle), the non-robust models even fall into the gray area of the plot where more confident decisions are likely incorrect. As expected, adversarially trained models not only make fewer mistakes in this case but are also better adjusted in terms of their confidence. Black-box attacks (Figure 1,top right) provide non-targeted out of domain samples. Adversarially trained models are overall better calibrated to this case, i.e. their mean confidences are hardly affected whereas non-robust models' confidences fluctuate heavily.

Four models stand out in Figure 1 (top left): two robust and two non-robust models which are much less confident in their true and false predictions than others. These less confident models are indeed trained from two different model architectures, with and without adversarial training. [59] uses a hypersphere embedding which normalizes the features in the intermediate layers

| Samples \ Robustness | Clean | PGD | Squares |
|---|---|---|---|
| non-robust models | 0.6736 ± 0.1208 | 0.6809 ± 0.1061 | 0.6635 ± 0.1156 |
| robust models | 0.1894 ± 0.1531 | 0.2688 ± 0.1733 | 0.2126 ± 0.1431 |

Table 1: Mean ECE (lower is better) and standard deviation over all non-robust model versus all their robust counterparts trained on CIFAR10. Robust model exhibit a significantly lower ECE on all samples.

and weights in the softmax layer, the other model [11] uses an ensemble of three different pretrained models (ResNet-50) to boost robustness. These architectural changes have a significant impact on the absolute model confidence, yet, do not necessarily lead to a better calibration. These models are under-confident in their correct predictions and tend to be comparably confident in wrong predictions.

Table 1 reports the mean ECE over all robust models and their non-robust counterparts. Robust models are better calibrated which results in a significantly lower ECE [3]. Figure 13 further visualizes the significant decrease in over-confidence of robust models w.r.t. their non-robust counterparts.

**CIFAR100**, although otherwise similar to CIFAR10, includes 100 classes and can be seen as a more challenging classification task. This is reflected in the reduced model accuracy on the clean and adversarial samples (Figure 1 , bottom). On this data, robust models are again less over-confident. They are slightly closer to the optimal calibration point in the lower right corner even on clean data and perform significantly better on PGD samples where the confidences of non-robust models are again reversed (middle). The Squares attack again illustrates the stable behavior of robust models' confidences[4]. We also report the ECE values for CIFAR100 in the Appendix. Please note that the accuracy of the CIFAR100 models is not very high (ranging between $56.87\%$ and $70.25\%$ even for clean samples), resulting in an unreliable calibration metric. Especially under PGD attacks, non-robust networks make mostly incorrect predictions such that the ECE collapses to being the expected confidence value of incorrect predictions (see eq. [1]), regardless of the confidences of the few correct predictions. In this case, ECE is not meaningful.

Another interesting observation is that non-robust models can achieve higher accuracy on the clean data and, quite surprisingly, on the applied black-box attacks (Figure 1, right). This indicates that most robust models overfit white-box attacks used during training and are not generalizing very well to other attacks. While making more mistakes, robust models still have a favorable distribution of confidence over non-robust models in this case.

**Model confidences can predict erroneous decisions.** Next, we evaluate the prediction confidences in terms of their ability to predict whether a network prediction is correct or incorrect. We visualize the ROC curves for all models and compare the averages of robust and non-robust models in Figure 3 (top row for CIFAR10, bottom row for CIFAR100), which allows us to draw conclusions about the confidence behavior. While robust and non-robust models perform on average very similarly on clean

---

[3]The models' full empirical confidence distributions are given in Figure 10 in the Appendix.

[4]The models' full empirical confidence distributions are given in Figure 11 in the Appendix

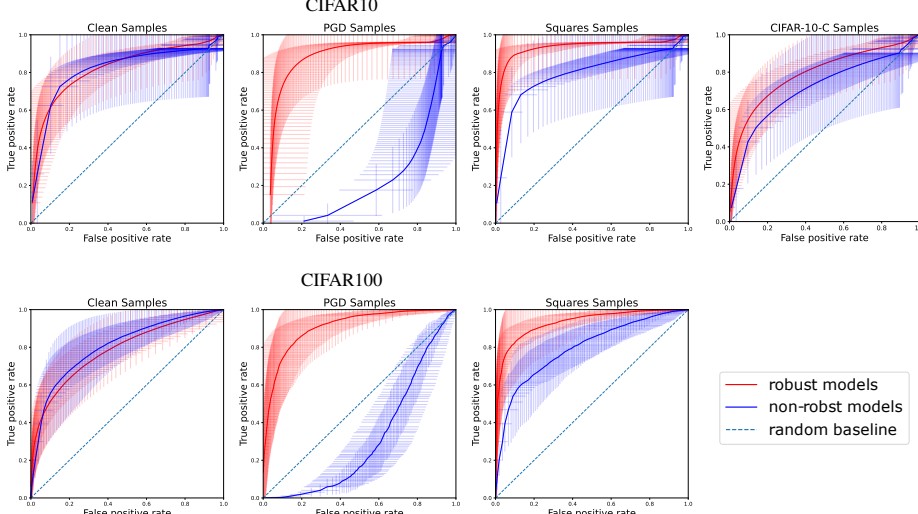

Figure 3: Average ROC curve for all robust and all non-robust models trained on CIFAR10 (top) and CIFAR100 (bottom). Standard deviation is marked by the error bars. The dashed line would mark a model which has the same confidence for each prediction. We observe that the models confidences can be an indicator for the correctness of the prediction. However, on PGD samples the non-robust models fail while the robust models can distinguish correct from incorrect predictions based on the prediction confidence.

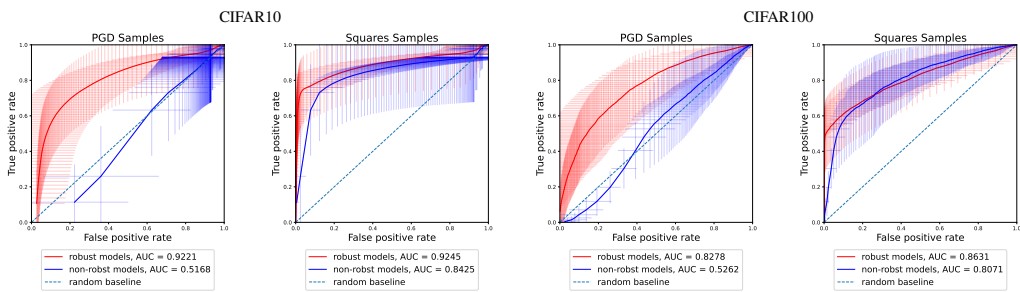

Figure 4: Average ROC curve over all robust and non-robust models of confidence on clean correctly classified samples and perturbated wrongly classified samples. The robust model confidences can be used as threshold for detection of white-box adversarial attacks (PGD). For black-box adversarial attacks (Squares) the robust as well as non-robust models can partially detect the erroneous samples.

data, robust model confidences can reliably predict erroneous classification results on adversarial examples where non-robust models fail. Also, for out-of-domain samples from the black-box attack *Squares* (middle right) and common corruptions [36] (right), robust models can reliably assess their prediction quality and can better predict whether their classification result is correct.

**Robust model confidences can detect adversarial samples.** Further, we evaluate the adversarial detection rate of the robust models based on their ROC curves (averaged over all robust models) in Figure 4, comparing the confidence of correct predictions on clean samples and incorrect predictions caused by adversarial attacks. We observe different behavior for gradient-based, white-box attacks and black-box attacks. While non-robust models fail completely against gradient based attacks they are almost as good as robust models for the detection of black-box attacks. Similarly, when taking the left two plots from Figure 3 into account, one might get the impression that non-robust models perform similar or even better on detecting erroneous samples compared to robust ones. Thus, we hypothesize that robust models indeed overfit the adversaries seen during training, as those are mostly gradient-based adversaries. Therefore we assume that adversarially trained models are not better calibrated in general, however, when strictly looking at overconfidence robust models are consistently less overconfident and therefore better applicable for safety critical applications.

**Downsampling techniques.** Most common CNNs apply downsampling to compress featuremaps with the intent to increase spatial invariance and overall higher sparsity. However, Grabinski et al. [29] showed that aliasing during the downsampling operation highly correlates with the lack of adversarial robustness, and provided a new downsampling operation, called *frequency low cut pooling* [28], which enables improved downsampling of the featuremaps. Figure 6 compares the confidence distribution of three different networks. The top row shows a PRN-18 baseline without adversarial training, the second row the approach by Grabinski et al. [28] applied to the same architecture (additional models are evaluated in the appendix D ), and the third row shows a robust model trained by Rebuffi et al. [62]. The baseline model is highly susceptible to adversarial attacks, especially under white-box attacks, while the two robust counter-parts remain low-confident in false predictions, and show higher confidence in correct predictions. However, while the model of Rebuffi et al. [62] shows a high variance amongst the predicted confidences, the approach by Grabinski et al. [28] significantly improves this by disentangling the confidences. Their model provides low-variance and high-confidence on correct predictions and reduced confidence on false predictions across all evaluated samples.

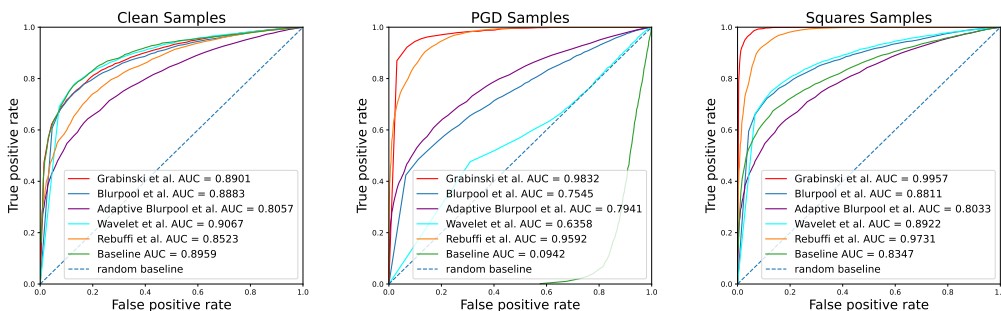

Figure 5: ROC curves and AUC values for different pooling variation in combination with adversarial training. FLC Pooling [28] outperforms all other pooling methods as well as the baseline.

In Figure 5, we compare different pooling methods combined with AT to standard pooling with AT as well as standard pooling without AT. The results show that the pooling method by Grabinski et al. [28] outperforms all other pooling methods. They consistently achieve the highest AUC under adversarial samples (white- and black-box attack) and are similar to the baseline on clean samples.

**Activation functions.** Next, we analyze the influence of activation functions. Only one Robust-Bench model utilizes an activation other than ReLU. Dai et al. [17] introduce learnable activation functions with the intent to improve robustness. Figure 7 shows at the top row a WRN-28-10 baseline model without AT, the model by Dai et al. [17] in the middle and a model with the same architecture adversarially trained by Carmon et al. [6].
Although this is an arguably sparse basis for a thorough investigation, we observe that the model by [17] can retain high confidence in correct predictions for both clean and perturbed samples. Furthermore, the model is much less confident in its wrong predictions for the clean as well as the adversarial samples. Similar to the used pooling variation, also the activation function seems to influence the model's calibration.

**Summary of low resolution datasets.** On CIFAR10 and CIFAR100 non-robust models can achieve higher standard accuracy and at least match or even exceed the performance of robust models under black-box attacks like *Squares*. Only under the white-box attack PGD, the robust models show higher accuracy. However non-robust models are highly over-confident in all their predictions and are hence limited in their applicability for real-world tasks. In contrast, the correctness of a robust models' prediction can be estimated by the prediction confidence. and is additionally serving as a defence against adversarial attacks. Further, we observe that the confidence of non-robust models decreases with increasing task complexity. In contrast, robust models are less affected by the increased task complexity and exhibit similar confidence characteristics on both datasets.

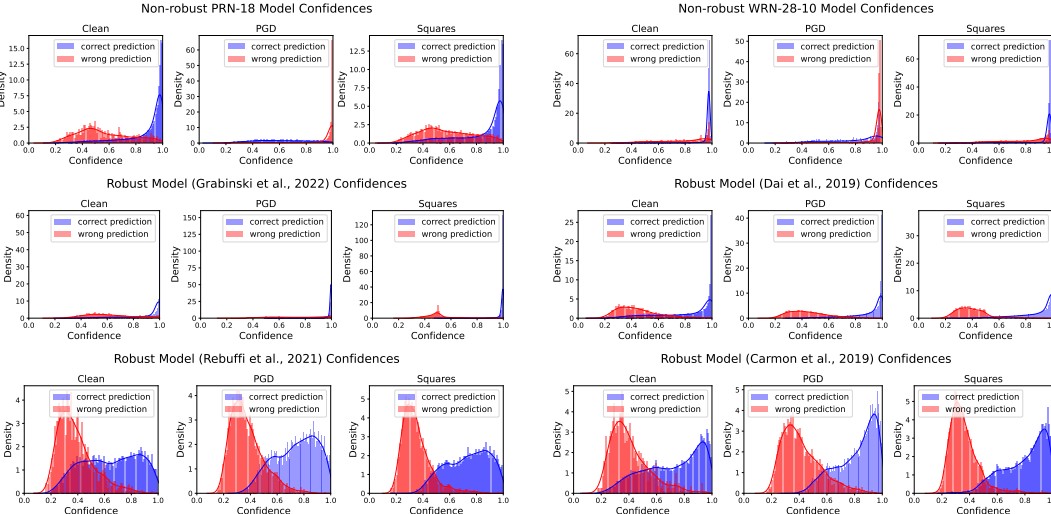

**Figure 6:** Confidence distribution on three different PRN-18. The first row shows a model without adversarial training and standard pooling, the second row the model by Grabinski et al. [28] which uses flc pooling instead of standard pooling and the third row shows the model by Rebuffi et al. [62] adversarially trained and with standard pooling.

**Figure 7:** Confidence distribution on three different WRN-28-10. The first row shows a model without adversarial training and standard activation (ReLU), the second row the model by Dai et al. [17] which uses learnable activation functions instead of fixed ones and the third row shows the model by Carmon et al. [6] adversarially trained and with the standard activation (ReLU).

## 3.3 ImageNet

We rely on the models provided by RobustBench [15] for our ImageNet evaluation. We report the clean and robust accuracy against *PGD* and *Squares* in Table 4 in the appendix. The non-robust model, trained without AT, achieves the highest performance on clean samples but collapses under white- and black-box attacks. Further, the models trained with multistep adversaries by Engstrom et al. [22] and Salman et al. [66] achieve higher robust and clean accuracy than the model trained by Wong et al. [83] which is trained with single-step adversaries. Moreover, the largest model, a WRN-50-2, yields the best robust performance. Still, the amount of robust networks on ImageNet is quite small, thus we can not make any generalized assumptions. Figure 9 shows the precision-recall curve for our evaluated models. Under evaluation with clean samples, the non-robust model without AT performs best. Under both attacks the largest model ( a WRN-50-2 by Salman et al. [66]) performs best and the worst performer is the smallest model (RN-18). This may be suggesting that bigger models can not only achieve the better trade-off in clean and robust accuracy but also more successfully disentangle confidences between correct and incorrect predictions. Figure 8 confirms that the over-confidence is decreased in robust models and the ECE is lower than in the non-robust models.

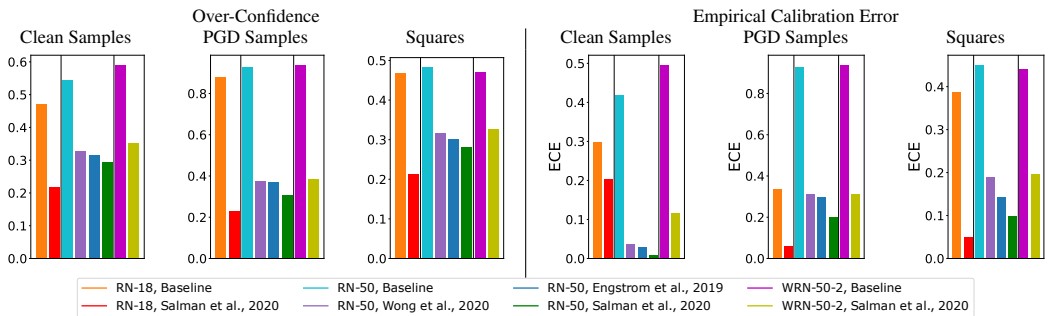

**Figure 8:** Overconfidence (left) and ECE (right) (lower is better) bar plots of the models trained on ImageNet provided by RobustBench [15] and their non-robust counterparts. The non-robust baselines exhibits the highest overconfidence and ECE. In contrast, the robust models are better calibrated.

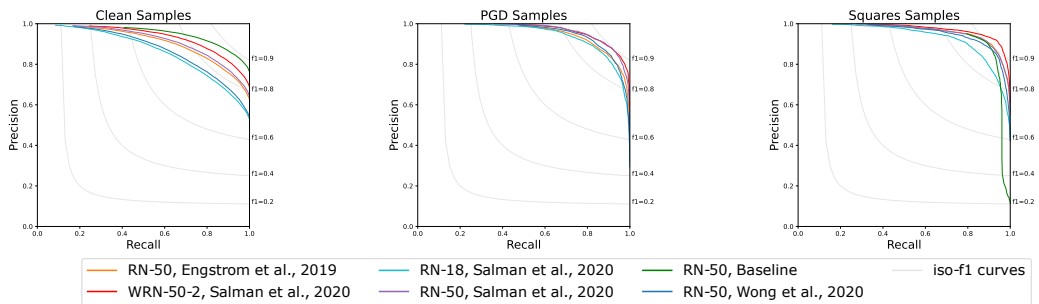

Figure 9: Precision Recall curves for the classification of correct versus erroneous predictions based on the confidence on ImageNet, evaluated over 10,000 samples. Robust and non-robust models are taken from RobustBench [15]. For clean samples (left) the non robust baseline performs best, while its confidences are less reliable under attack (middle and right). The robust WRN-50-2 by Salman et al. [66] performs best on the PGD and Squares samples.

## 3.4 Discussion

Our experiments confirm that the prediction confidences of non-robust models are highly over-confident, especially under gradient based, white-box attacks. However, when confronted with clean samples, common corruptions or unseen black-box attacks like Squares [4] non-robust and robust models are equally able to detect wrongly classified samples based on their prediction confidence. Indicating that adversarially trained networks overfit the kind of adversaries seen during training.

Further, our results indicate that the selection of the activation functions as well as the downsampling are important factors for the models' performance and confidence. The method by Grabinski et al. [28], which improves the downsampling, as well as the method by Dai et al. [17], which improves the activation function, exhibit the best calibration for the networks prediction; High confidence on correct predictions and low confidence on the incorrect ones. While further optimizing deep neural networks' architectures and training schemes, we should consider the synopsis of model robustness and calibration instead of optimizing each of these aspects separately.

**Limitations.** Our evaluation is based on the models provided on RobustBench [15]. Thus the amount of networks on more complex datasets, like ImageNet, is rather small and therefore the evaluation not universally applicable. While the number of models for CIFAR is large, the proposed database can only be understood as a starting point for future research. This is particularly true for the analysis of neural network building blocks - models that are adversarially trained and employ smooth activation functions might be very promising concerning their calibration but a more in-depth analysis of this setting with new, dedicated datasets is desirable. Additionally, we rely simply on the confidence obtained after the Softmax layer, while there are many other metrics for uncertainty measurement.

## 4 Conclusion

We provide an extensive study on the confidences of robust models and observe an overall trend: robust models tent to be less over-confident than non-robust models. Thus, while achieving a higher robust accuracy, adversarial training generates models that are less overconfident. Further, the prediction confidence of robust models can actually be used to reject wrongly classified samples on clean data and even adversarial examples.

Moreover, we see indications that exchanging simple building blocks like the activation function [17] or the downsampling method [28] alters the properties of robust models with respect to confidence calibration. On the examples we investigate, the models' prediction confidence on their correct predictions can be increased while the confidence on the erroneous predictions remains low. Our findings should nurture future research on jointly considering model calibration and robustness.

However, robust models' overall performance on robustness tasks are highly questionable as they seem to overfit the adversaries seen during training.

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
