## A Non-robust Model Training

For training, *CIFAR-10/100* data was zero-padded by 4 px along each dimension, and then transformed using $32 \times 32$ px random crops, and random horizontal flips. Channel-wise normalization was replicated as reported by the original dataset authors. Training hyper parameters have been set to an initial learning rate of 1e-2, a weight decay of 1e-2, a batch-size of 256 and a nesterov momentum of 0.9. We scheduled the SGD optimizer to decrease the learning rate every 30 epochs by a factor of $\gamma = 0.1$ and trained for a total of 125 epochs. The loss is determined using Categorical Cross Entropy and we used the model obtained at the epoch with the highest validation accuracy. Training was executed on a *A+ Server* SYS-2123GQ-NART-2U machine with four *NVIDIA* A100-SXM4-40GB GPUs for approximately 17 GPU hours. Training *ImageNet1k* architectures with our hyperparameters resulted in a rather poor performance and we therefore rely on the baseline model without AT provided by *timm* [82].

## B Additional Evaluation CIFAR10/100

In this section we provide an overview over ECE on CIFAR10 and CIFAR100 of all robust models and their non-robust conunterparts.

### B.1 Confidence Distribution

The model confidence distributions are shown in Figure 10 and Figure 11. Each row contains the robust and non-robust counterpart and their confidence distributions on the clean samples and the perturbated samples by PGD and Squares.

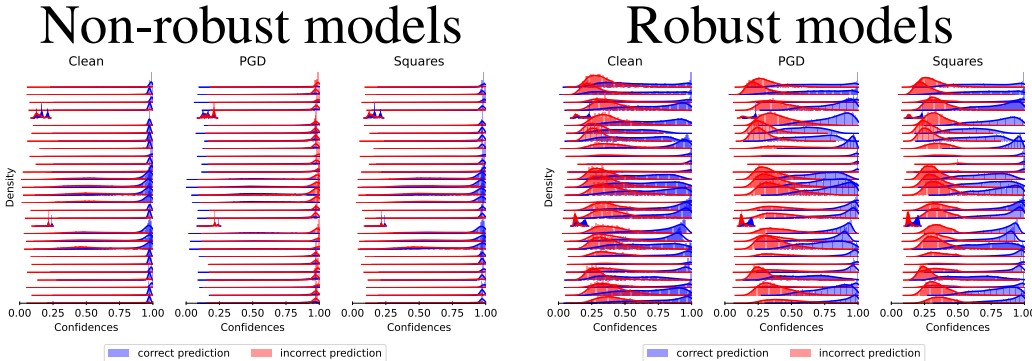

Figure 10: Density plots for robust and non-robust models on CIFAR10 over the models confidence on its correct and incorrect predictions. Each row contains the same model adversarially and standard trained. The non-robust models show high confidence in all of their predictions, however, those might be wrong. Especially in the case of PGD samples, the models are highly confident in their false predictions. In contrast, the robust models are better calibrated. The robust models are confident in their correct predictions and less confident in their false predictions.

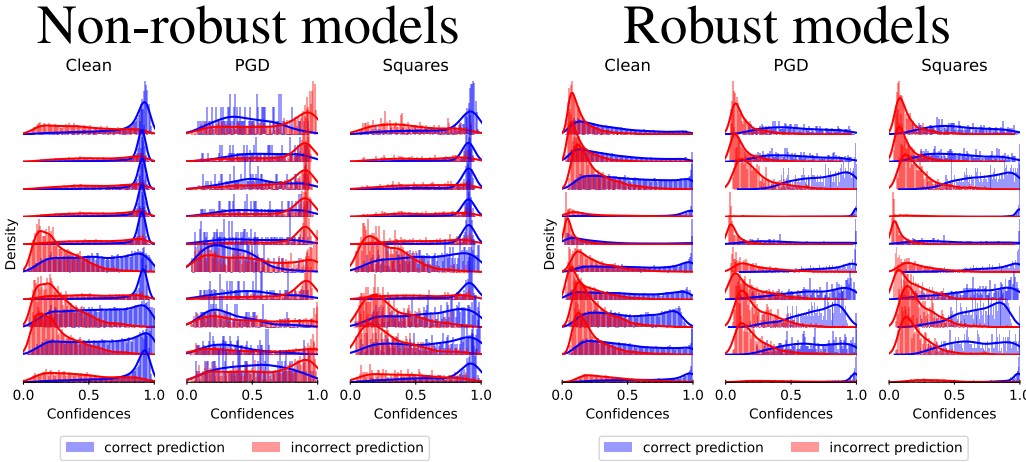

Figure 11: Density plots for robust and non-robust models on CIFAR100 over the models confidence on its correct and incorrect predictions. Each row contains the same model adversarially and standard trained. The non-robust models show high confidence in all of their predictions, however, those might be wrong. Especially in the case of PGD samples, the models are highly confident in their false predictions. In contrast, the robust models are better calibrated. The robust models are confident in their correct predictions and less confident in their false predictions.

## B.2 Overconfidence and ECE

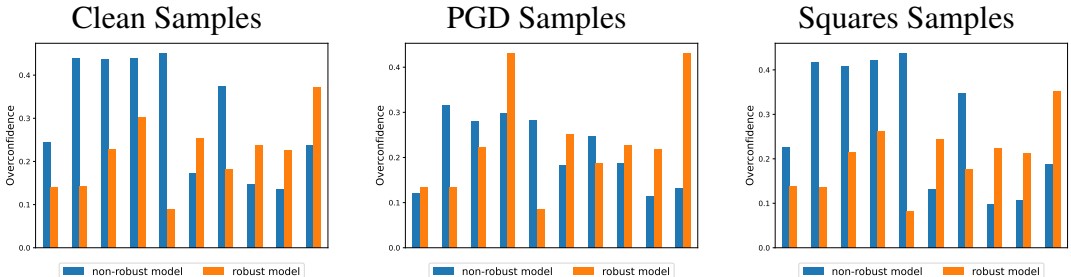

Figure 12: Overconfidence (lower is better) bar plots of robust models and their non-robust counterparts trained on CIFAR100.

Similar, the confidence distributions for the robust and non-robust counterparts on CIFAR100 are depicted in Figure 11.

| Samples \ Robustness | Clean | PGD | Squares |
|---|---|---|---|
| non-robust models | $0.3077 \pm 0.1257$ | $0.2159 \pm 0.0738$ | $0.2780 \pm 0.1348$ |
| robust models | $0.2962 \pm 0.1722$ | $0.2307 \pm 0.1494$ | $0.2076 \pm 0.1247$ |

Table 2: Mean ECE (lower is better) and standard deviation over all non-robust model versus all their robust counterparts trained on CIFAR100. Robust model exhibit a significantly lower ECE on all samples.

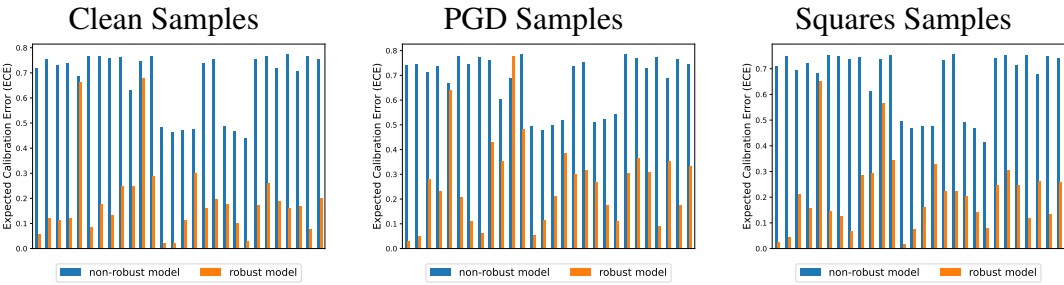

Figure 13: ECE (lower is better) bar plots of robust models and their non-robust counterparts trained on CIFAR10.

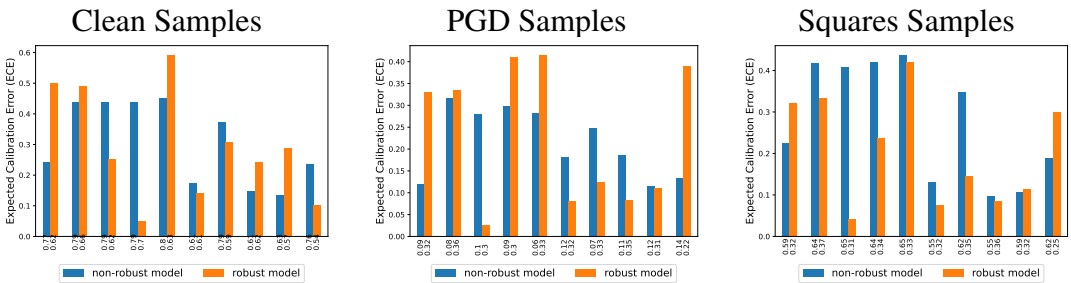

Figure 14: ECE (lower is better) bar plots of robust models and their non-robust counterparts trained on CIFAR100. The models accuracy are marked for the different samples for each bar.

## B.3   Precision Recall

For completeness, we included the Precision Recall curves on CIFAR10 and CIFAR100 as mean over all robust and non-robust models with marked standard deviation.

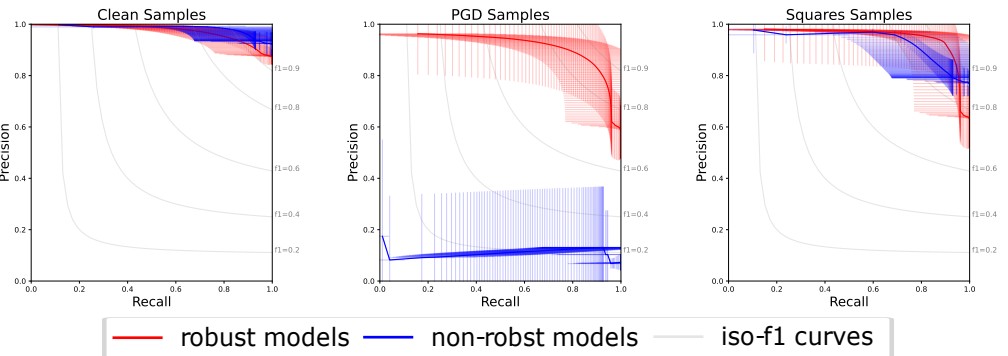

Figure 15: Average precision recall curve for all robust and all non-robust models trained on CIFAR10. Standard deviation is marked by the error bars. For the clean samples, the non-robust models can distinguish slightly better in correct and incorrect predictions based on the confidence of the prediction. The superior of the robust models are visible on the samples created by PGD, the non-robust models are not able to distinguish. However, for the samples created by Squares the classification into correct and incorrect predictions based on the confidence is almost equally possible for robust and non-robust models.

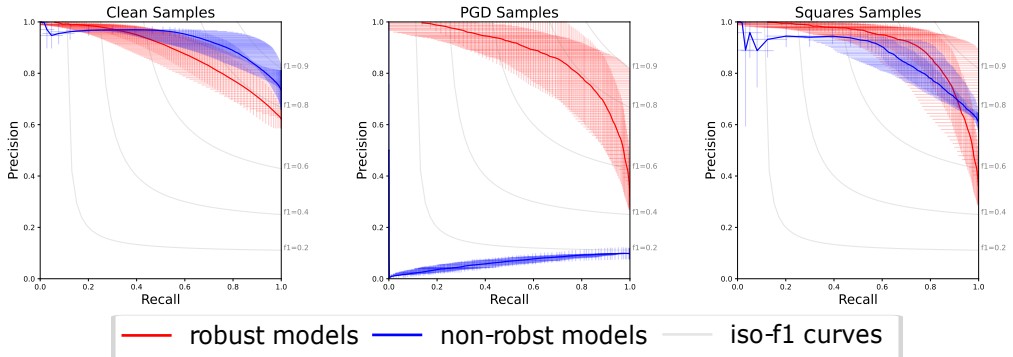

Figure 16: Average precision recall curve for all robust and all non-robust models trained on CIFAR100 for 1000 samples. Standard deviation is marked by the error bars. For the clean samples, the non-robust models can distinguish slightly better in correct and incorrect predictions based on the confidence of the prediction. The superior of the robust models are clearly visible on the samples created by PGD, the non-robust models are not able to distinguish. However, for the samples created by Squares the classification into correct and incorrect predictions based on the confidence is almost equally possible for robust and non-robust models.

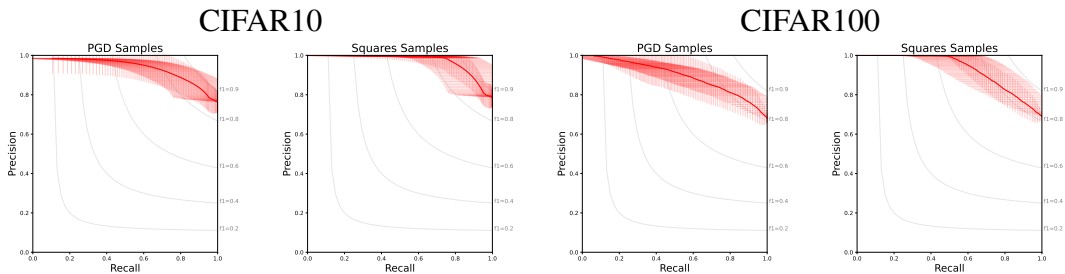

Figure 17: Precision Recall curve between confidence of clean correct samples and perturbated wrong samples on CIFAR10 and CIFAR100. The robust model confidences can be used as threshold for detection of adversarial attacks.

## C  CIFAR10-C Evaluation

Additionally to the previously studied attacks, we evaluate the confidence of robust versus non-robust CIFAR-10 models on the out-of-distribution dataset CIFAR10-C with severity level 4 (although the results for other severity levels follow the same trajectory and are omitted). There we benchmark models robust to adversarial attacks and their non-robust counterparts and evaluate the prediction confidence.

### C.1  Overconfidence

First, we compare the models overconfidence with respect to each corruption type. In accordance with our findings on adversarial perturbations, robust models are much less overconfident than their non-robust counterparts. Figure 18 shows the overconfidence of each model pair for each corruption type. We can clearly see that robust models are generally much less overconfident.

### C.2  ROC-curve

Regarding the mean ROC-curves (Figure 19) our results show that robust models tend to be better calibrated than non-robust models. However, robust models are inferior with respect to their calibration on corruptions changing the color palette of the image, like fog, brightness, contrast, and saturation.

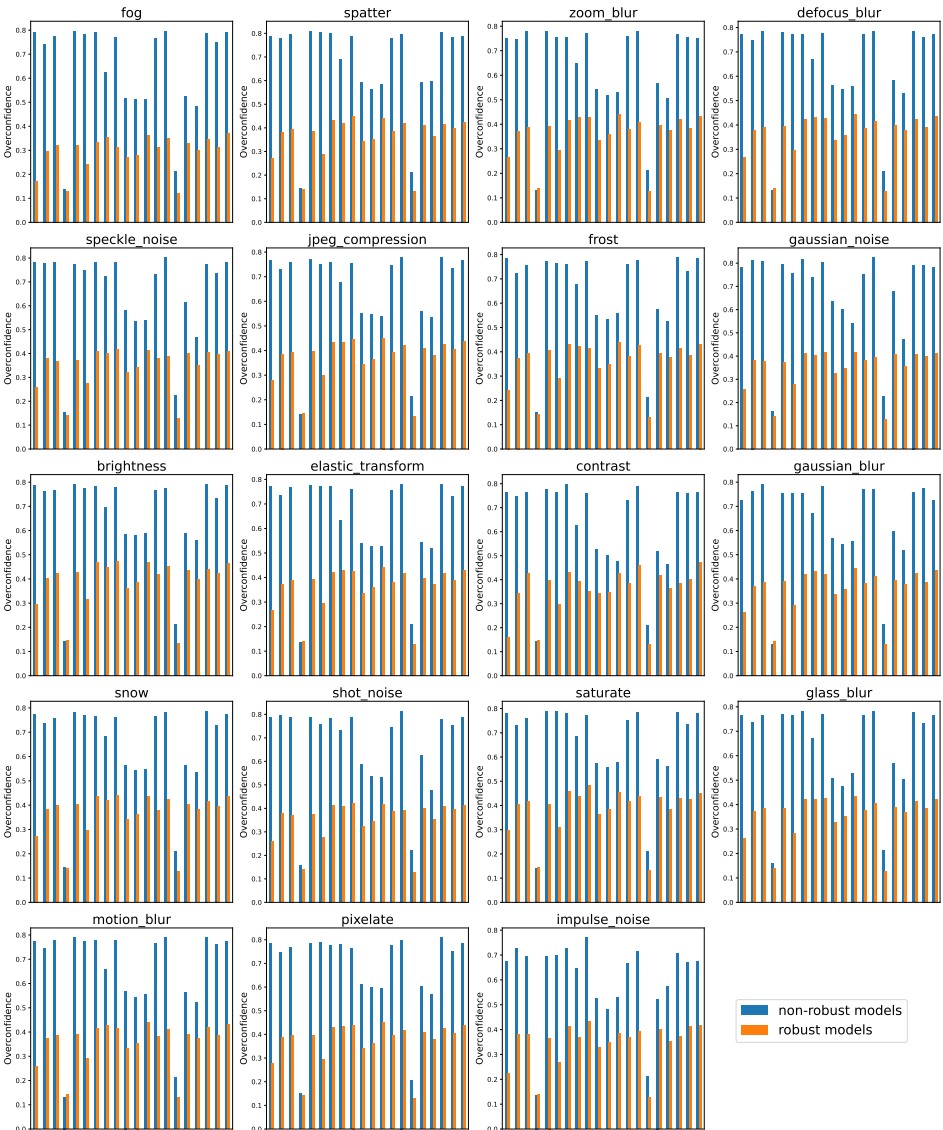

Figure 18: Overconfidence for each robust CIFAR-10 model and the respective normal counterpart evaluated on CIFAR10-C.

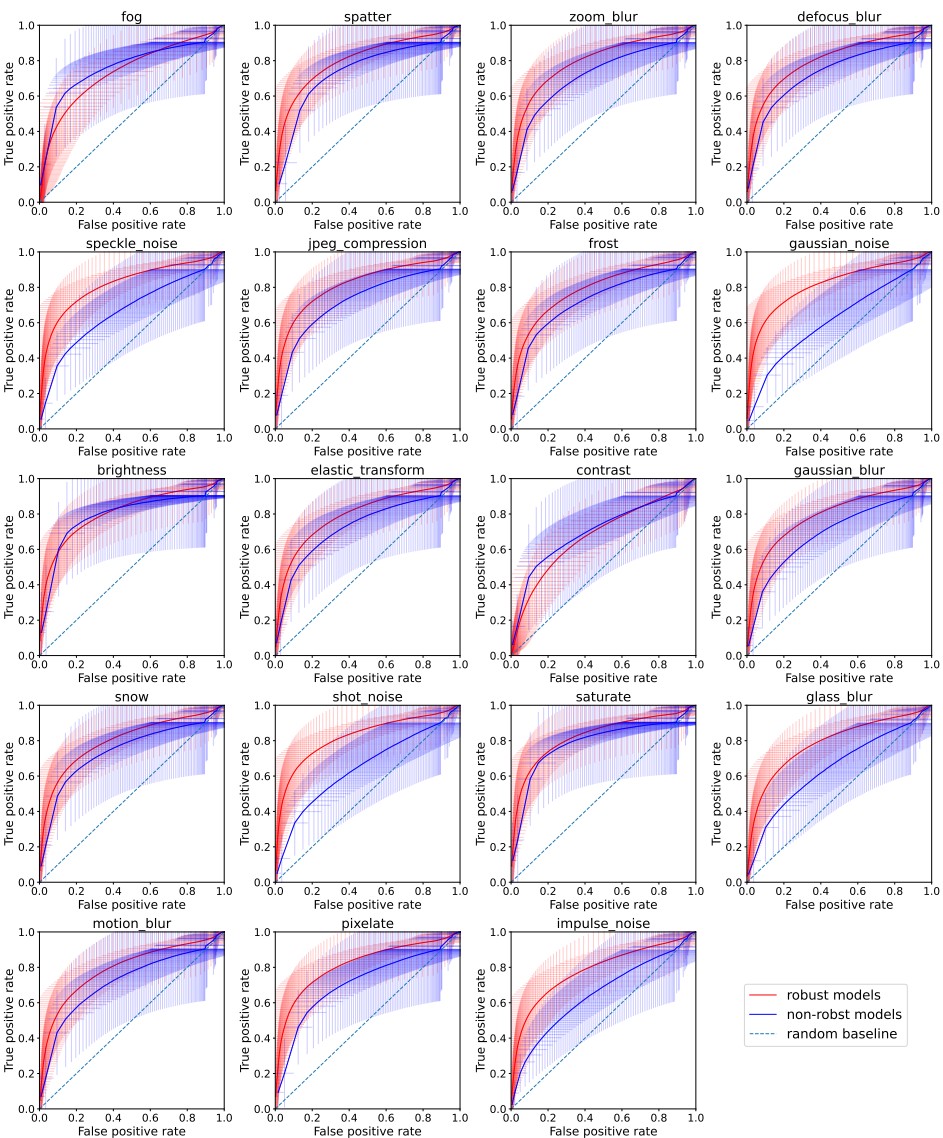

Figure 19: Mean ROC curves for each robust and non-robust CIFAR-10 model pair evaluated on CIFAR10-C.

## D FLC Pooling

We evaluate different robust PRN-18 networks trained with flc pooling [28] and FGSM AT in terms of their confidence distribution. For training, we used the training script provided by [83]. We trained with ten different seeds and run for 300 epochs, choosing the batchsize to be 128, a momentum of 0.9, weight decay of 0.0005, a cycling learning rate with minimum value of 0 and maximum value of 0.2, for the adversarial samples we used FGSM with an $\epsilon$ of $8/255$ and $\alpha$ of $10/255$. Figure 20 shows the confidence distribution over all ten models and the standard deviation between those models. We can observe that the models with flc pooling are able to disentangle the correct from the incorrect prediction by the prediction confidence. The models provide low-variance and high-confidence in correct predictions and reduced confidence in false predictions across all evaluated samples.

## E Downsampling and Activation

### E.1 AUC

To show the impact of improved downsampling and activation functions we provide the ROC curves and AUC values of the models with and without those improved building blocks (similar to Figure 20 and Figure 7). Figure 21 shows the ROC curves on the improved building blocks as well as on comparable robust models with the same architecture. One can see that the improved building blocks results in slightly better calibration. The corresponding AUC values are reported in Table 3.

### E.2 CIFAR10-C

Next, we compare the confidence impact of improved downsampling operations and activations on out-of-distribution data. Here we summarize our findings by the mean over all corruptions. Figure 22 shows that robust models are on average better calibrated than normal models. The impact of improved downsampling or activation functions is marginal.

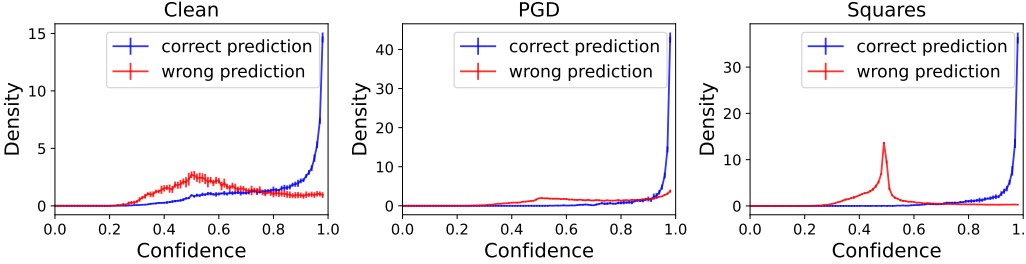

Figure 20: Additional confidence distribution evaluation over ten models (PRN-18) trained on CIFAR10 with flc pooling [28] and AT FGSM [83]. We used 100 bins and present the mean and standard deviation of the ten different models for each bin.

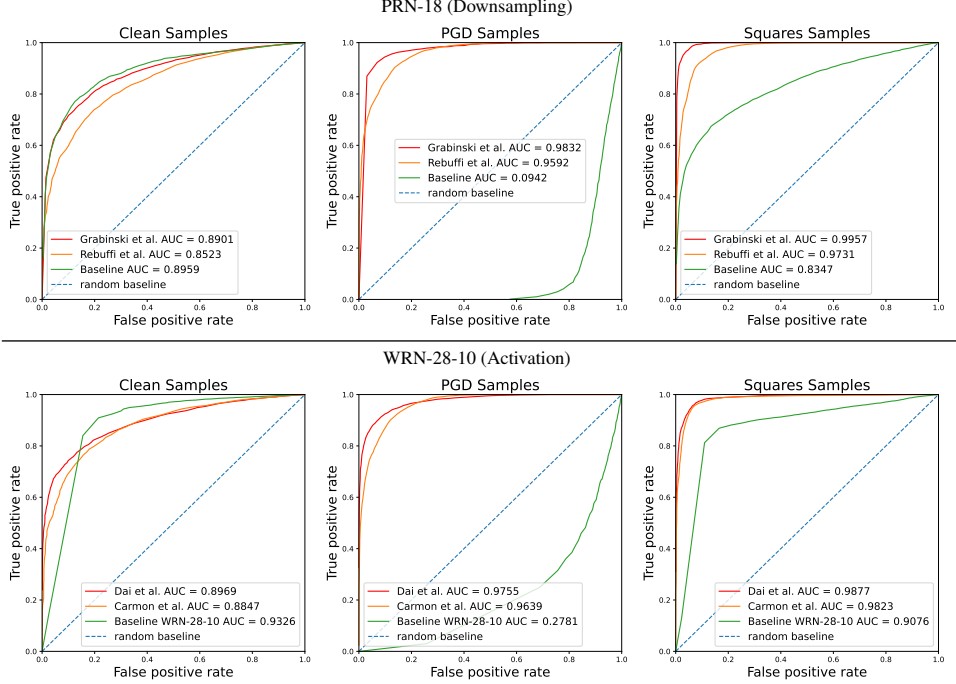

Figure 21: ROC curves for robust models with and without special building block, like downsampling (top) and activation (bottom).

| Robust Model | Clean | PGD | Squares |
|---|---|---|---|
| Baseline PRN-18 | 0.8958 | 0.0942 | 0.8347 |
| Grabinski et al. [28] | 0.8901 | 0.9832 | 0.9923 |
| Rebuffi et al. [62] | 0.8523 | 0.9592 | 0.9731 |
| Baseline WRN-28-10 | 0.9326 | 0.2781 | 0.9076 |
| Dai et al. [17] | 0.8969 | 0.9755 | 0.9877 |
| Carmon et al. [6] | 0.8847 | 0.9639 | 0.9823 |

Table 3: AUC value for the ROC curves of different robust models provided by [6, 17, 28, 62].

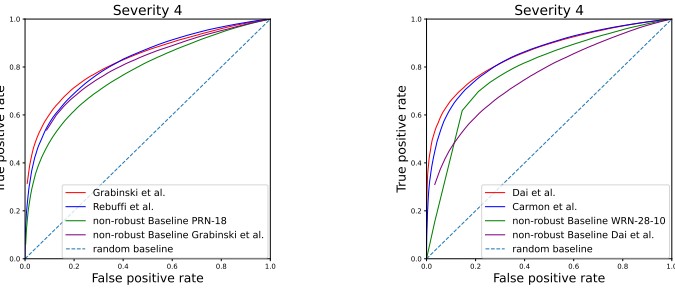

Figure 22: ROC curve for improved downsampling (left) and activation function (right) on CIFAR10-C corruptions. Robust models are superior to the normal models, and, the impact of activation and pooling is marginal.

# F Additional Evaluation on ImageNet

Table 4 reports the accuracy evaluation of the robust models as well as the baseline on ImageNet. The accuracy is reported on the clean as well as on the perturbated samples by PGD and Squares with an $\epsilon$ of $4/255$.

| Method | Architecture | Clean Acc ↑ | PGD Acc ↑ | Squares Acc ↑ |
|---|---|---|---|---|
| Baseline | RN50 | **76.13** | 0.00 | 11.48 |
| Engstrom et al. [22] | RN50 | 62.41 | 35.47 | 54.93 |
| Wong et al. [83] | RN50 | 53.83 | 29.43 | 42.26 |
| Salman et al. [66] | RN50 | 63.87 | 42.23 | 56.58 |
| Salman et al. [66] | WRN50-2 | 68.41 | **44.75** | **61.29** |
| Salman et al. [66] | RN18 | 52.50 | 31.92 | 43.81 |

Table 4: Clean and robust accuracy against PGD and Squares (higher is better) over 10000 samples.

For completeness, we included the ROC curve on the clean as well as the perturbated samples for the robust models and the baseline on ImagNet in figure 23.

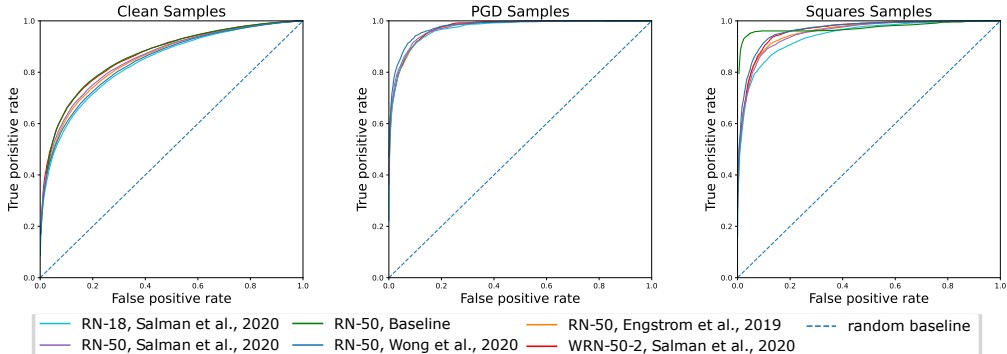

Figure 23: ROC curves for the robust models and the non-robust baseline trained on ImageNet provided on RobustBench [15].

# G  Model Overview

The robust checkpoints provided by *RobustBench* [15] are licensed under the MIT Licence. The clean models for ImageNet are provided by *timm* [82] under the Apache 2.0 licence.

| Paper | Dataset | Architecture | Adv. Trained Clean Acc. | Adv. Trained Robust Acc. | Norm. Trained Clean Acc. | Norm. Trained Robust Acc. |
|---|---|---|---|---|---|---|
| [3] | cifar10 | PreActResNet-18 | 79.84 | 43.93 | 94.51 | 0.0 |
| [6] | cifar10 | WideResNet-28-10 | 89.69 | 59.53 | 95.10 | 0.0 |
| [67] | cifar10 | WideResNet-28-10 | 88.98 | 57.14 | 95.10 | 0.0 |
| [80] | cifar10 | WideResNet-28-10 | 87.50 | 56.29 | 95.10 | 0.0 |
| [38] | cifar10 | WideResNet-28-10 | 87.11 | 54.92 | 95.35 | 0.0 |
| [64] | cifar10 | WideResNet-34-20 | 85.34 | 53.42 | 95.46 | 0.0 |
| [87] | cifar10 | WideResNet-34-10 | 84.92 | 53.08 | 95.26 | 0.0 |
| [22] | cifar10 | ResNet-50 | 87.03 | 49.25 | 94.90 | 0.0 |
| [11] | cifar10 | ResNet-50 | 86.04 | 51.56 | 86.50 | 0.0 |
| [41] | cifar10 | WideResNet-34-10 | 83.48 | 53.34 | 95.26 | 0.0 |
| [59] | cifar10 | WideResNet-34-20 | 85.14 | 53.74 | 76.30 | 0.0 |
| [83] | cifar10 | PreActResNet-18 | 83.34 | 43.21 | 94.25 | 0.0 |
| [21] | cifar10 | WideResNet-28-4 | 84.36 | 41.44 | 94.33 | 0.0 |
| [86] | cifar10 | WideResNet-34-10 | 87.20 | 44.83 | 95.26 | 0.0 |
| [89] | cifar10 | WideResNet-34-10 | 84.52 | 53.51 | 95.26 | 0.0 |
| [84] | cifar10 | WideResNet-28-10 | 88.25 | 60.04 | 95.10 | 0.0 |
| [84] | cifar10 | WideResNet-34-10 | 85.36 | 56.17 | 95.64 | 0.0 |
| [26] | cifar10 | WideResNet-70-16 | 85.29 | 57.20 | 87.91 | 0.0 |
| [26] | cifar10 | WideResNet-70-16 | 91.10 | 65.88 | 87.91 | 0.0 |
| [26] | cifar10 | WideResNet-34-20 | 85.64 | 56.86 | 88.33 | 0.0 |
| [26] | cifar10 | WideResNet-28-10 | 89.48 | 62.80 | 88.20 | 0.0 |
| [68] | cifar10 | WideResNet-34-10 | 85.85 | 59.09 | 95.64 | 0.0 |
| [68] | cifar10 | ResNet-18 | 84.38 | 54.43 | 94.87 | 0.0 |
| [70] | cifar10 | WideResNet-34-10 | 86.84 | 50.72 | 95.26 | 0.0 |
| [9] | cifar10 | WideResNet-34-10 | 85.32 | 51.12 | 95.35 | 0.0 |
| [16] | cifar10 | WideResNet-34-20 | 88.70 | 53.57 | 95.44 | 0.0 |
| [16] | cifar10 | WideResNet-34-10 | 88.22 | 52.86 | 95.26 | 0.0 |
| [90] | cifar10 | WideResNet-28-10 | 89.36 | 59.64 | 95.10 | 0.0 |
| [62] | cifar10 | WideResNet-28-10 | 87.33 | 60.75 | 88.20 | 0.0 |
| [62] | cifar10 | WideResNet-106-16 | 88.50 | 64.64 | 86.92 | 0.0 |
| [62] | cifar10 | WideResNet-70-16 | 88.54 | 64.25 | 87.91 | 0.0 |
| [62] | cifar10 | WideResNet-70-16 | 92.23 | 66.58 | 87.91 | 0.0 |
| [71] | cifar10 | WideResNet-28-10 | 89.46 | 59.66 | 95.10 | 0.0 |
| [71] | cifar10 | WideResNet-34-15 | 86.53 | 60.41 | 95.50 | 0.0 |
| [62] | cifar10 | PreActResNet-18 | 83.53 | 56.66 | 89.01 | 0.0 |
| [61] | cifar10 | PreActResNet-18 | 89.02 | 57.67 | 89.01 | 0.0 |
| [61] | cifar10 | PreActResNet-18 | 86.86 | 57.09 | 89.01 | 0.0 |
| [61] | cifar10 | WideResNet-34-10 | 91.47 | 62.83 | 88.67 | 0.0 |
| [61] | cifar10 | WideResNet-28-10 | 88.16 | 60.97 | 88.20 | 0.0 |
| [40] | cifar10 | WideResNet-34-R | 90.56 | 61.56 | 95.60 | 0.0 |
| [40] | cifar10 | WideResNet-34-R | 91.23 | 62.54 | 95.60 | 0.0 |
| [1] | cifar10 | ResNet-18 | 80.24 | 51.06 | 94.87 | 0.0 |
| [1] | cifar10 | WideResNet-34-10 | 85.32 | 58.04 | 95.26 | 0.0 |
| [27] | cifar10 | WideResNet-70-16 | 88.74 | 66.11 | 87.91 | 0.0 |
| [17] | cifar10 | WideResNet-28-10-PSSiLU | 87.02 | 61.55 | 85.53 | 0.0 |
| [27] | cifar10 | WideResNet-28-10 | 87.50 | 63.44 | 88.20 | 0.0 |

| Paper | Dataset | Architecture | Adv. Trained Clean Acc. | Adv. Trained Robust Acc. | Norm. Trained Clean Acc. | Norm. Trained Robust Acc. |
|---|---|---|---|---|---|---|
| [27] | cifar10 | PreActResNet-18 | 87.35 | 58.63 | 89.01 | 0.0 |
| [8] | cifar10 | WideResNet-34-10 | 85.21 | 56.94 | 95.64 | 0.0 |
| [8] | cifar10 | WideResNet-34-20 | 86.03 | 57.71 | 95.29 | 0.0 |
| [26] | cifar100 | WideResNet-70-16 | 60.86 | 30.03 | 60.56 | 0.0 |
| [26] | cifar100 | WideResNet-70-16 | 69.15 | 36.88 | 60.56 | 0.0 |
| [16] | cifar100 | WideResNet-34-20 | 62.55 | 30.20 | 80.46 | 0.0 |
| [16] | cifar100 | WideResNet-34-10 | 70.25 | 27.16 | 79.11 | 0.0 |
| [16] | cifar100 | WideResNet-34-10 | 60.64 | 29.33 | 79.11 | 0.0 |
| [9] | cifar100 | WideResNet-34-10 | 62.15 | 26.94 | 78.75 | 0.0 |
| [84] | cifar100 | WideResNet-34-10 | 60.38 | 28.86 | 78.79 | 0.0 |
| [70] | cifar100 | WideResNet-34-10 | 62.82 | 24.57 | 79.11 | 0.0 |
| [38] | cifar100 | WideResNet-28-10 | 59.23 | 28.42 | 79.16 | 0.0 |
| [64] | cifar100 | PreActResNet-18 | 53.83 | 18.95 | 76.18 | 0.0 |
| [62] | cifar100 | WideResNet-70-16 | 63.56 | 34.64 | 60.56 | 0.0 |
| [62] | cifar100 | WideResNet-28-10 | 62.41 | 32.06 | 61.46 | 0.0 |
| [61] | cifar100 | PreActResNet-18 | 56.87 | 28.50 | 63.45 | 0.0 |
| [61] | cifar100 | PreActResNet-18 | 61.50 | 28.88 | 63.45 | 0.0 |
| [1] | cifar100 | PreActResNet-18 | 62.02 | 27.14 | 76.66 | 0.0 |
| [1] | cifar100 | WideResNet-34-10 | 65.73 | 30.35 | 79.11 | 0.0 |
| [8] | cifar100 | WideResNet-34-10 | 64.07 | 30.59 | 79.11 | 0.0 |
| [83] | imagenet | ResNet-50 | 55.62 | 26.24 | 80.37 | 0.0 |
| [22] | imagenet | ResNet-50 | 62.56 | 29.22 | 80.37 | 0.0 |
| [66] | imagenet | ResNet-50 | 64.02 | 34.96 | 80.37 | 0.0 |
| [66] | imagenet | ResNet-18 | 52.92 | 25.32 | 69.74 | 0.0 |
| [66] | imagenet | WideResNet-50-2 | 68.46 | 38.14 | 81.45 | 0.0 |