# OpenReview forum: "Robust Models are less Over-Confident"
_NeurIPS.cc/2022/Conference — NeurIPS 2022 Accept_

### Official Review · Reviewer_aQN4 · 2022-07-10

**Rating:** 6
**Confidence:** 5
**Soundness:** 3 good
**Presentation:** 3 good
**Contribution:** 3 good

**Summary:**

This paper focuses on the problem of CNNs being overconfident with their predictions and the effect of adversarial training on this matter. It contains extensive empirical analyses of model confidence scores:
- Adversarially trained (AT) robust models.
- Model architectures with parametrized activation functions and downsampling layers (as explored in [13]).

The authors show:
- AT results in more calibrated models. They do so by:
1. Taking existing robust model checkpoints from [15].
2. Train an identical model that's not trained with AT.
3. Creating a validation set of clean and adversarial samples in white-box and black-box settings.
4. They support their claims on CIFAR-10 and -100 and ImageNet datasets. They visualize correct and incorrect class confidence scores, predicted score distribution, and the expected calibration error metric.

- The robust models are less confident on attacked samples. This has been shown by comparing the ROC of robust vs. non-robust models on clean and attacked samples. They also show that the robust model confidences can be used to detect adversarial examples directly.

- They show that improved building blocks result in lower confidence scores on adversarially attacked samples. They visualize the distribution of confidence scores when these modifications are made.



**Questions:**

-  In Figures 5 and 6, it seems that the ROC AUC of the second row can be good for attack detection. Can you please report this value? It can be the case that AT and learnable activation functions or FLC pooling can result in similar AUC. This could mean that in case of calibration is not important and we just care about attack detection, one can rely on swapping the building blocks which have lower complexity compared to AT.

- Is this a fair conclusion: learnable activation blocks and FLC generalize better to unseen attacks compared to AT for attack detection How do Figures 4, 5, and 6 look like on unseen attacks?

- typo in Figure 7: rigth -> right.

**Limitations:**

The authors have adequately addressed the limitations of their work.

**Strengths And Weaknesses:**

Strengths:
- Highlighting an important property of adversarial training.
- Extensive empirical analysis covering different aspects of their hypothesis.
- Paper writing and organization.

Weaknesses:
- In Figure 4, I would expect to see the ROC of clean models used for the same purpose although potentially it's not great.
- More quantitative metrics could be reported: (a) The ROC AUC for attack detection of different approaches for experiments in Figures 4, 5, and 6, 8 for robust and non-robust models and the different building blocks. ECE for the experiments is interesting to see. I couldn't find these values in the appendix either.
- There could be more experiments focused on the generalization of claims to unseen attacks.

---

> ### Author Response · Authors · 2022-08-02
> **Response to Review**
>
> Thank you for your time and effort you put into the review of our paper. We address the points listed under weaknesses and questions one-by-one in the order they appear in the review.
>
> W1: [ROC of clean models in Figure 4] Thank you for the suggestion, we added non-robust models into Figure 4. We could observe that those models indeed fail to recognize PGD samples. They are able to distinguish clean from Squares samples quite well.
>
> W2: [More quantitative metrics] We report the density plots of all models in the appendix in Figures 9 and 10. There one can see that almost all models show similar calibrations except for two models which are described from line 199 to line 204 in the manuscript (202 to 205 in the revised manuscript). The ECE for the different models are reported in the appendix Figure 12 and Figure 13. Due to the amount of models we only reported the values without each specific name of the model. Figure 8 where we show the Precision-Recall Curve for ImageNet, the equivalent ROC curve is reported in the appendix Figure 17 (revised manuscript Figure 22). Further, we report the Precision-Recall curves for CIFAR10 and CIFAR100 in the appendix Figure 14 and Figure 15. We tried to restructure the appendix for better clarity.
> Additionally in our revised manuscript, we added an evaluation on the improved downsampling and activation by inspecting the ROC curves and AUC values for these models and their comparable models in detail in Figure 20 and Table 3 in the appendix.
>
> W3: [Unseen attacks] Please note that the Squares attack is an unseen attack during training for both robust and non-robust models. To further strengthen our evaluation in this respect, we additionally evaluate CIFAR10-C as a generalization task on the robust model and their non-robust counterparts. CIFAR10-C is a dataset with common corruptions and therefore usually allows to make observations on model behavior in unseen scenarios.  We observe a similar trend as in the adversarial samples. Robust models are less over-confident. The full evaluation is now included in our revised manuscript (Section C).
>
> Q1: [ROC AUC for attack detection in Figures 5 and 6] FLC, as well as the learned activation functions, use AT to achieve robustness, thus only swapping the building blocks leads not to an increase in robustness, however, the disentanglement of the confidence score is better calibrated for those two approaches. However, when comparing the AUC for each ROC curve we can see that the improved building blocks lead to higher AUC. We added the full results in the appendix of the revision of our paper (Table 3).
>
> Q2: [“learnable activation blocks and FLC generalize better to unseen attacks compared to AT”,  Unseen attacks]
> From our results, it can not be concluded that learnable activation blocks or FLC generalize better than AT, because both models are additionally trained with AT. We can only conclude that FLC or learnable activations can have an additional positive impact.
> We used the black-box attack Squares to evaluate against unseen attacks. Specifically, none of the models has seen Squares samples during training. Further, the FLC pooling is trained with simple FGSM thus the PGD samples are also unseen for this model. However, the model including learned activation functions is trained with PGD and thus has seen PGD samples already during training. Squares samples are out-of-domain.
>
> In summary, we incorporate your suggestions into our revised manuscript as follows:
> - We included the non-robust models in Figure 4.
> - We restructured the appendix for more clarity.
> - To further strengthen our evaluation for unseen domain shifts, we additionally evaluate CIFAR10-C as a generalization task on the robust model and their non-robust counterparts and observe a similar trend as on the adversarial samples. Robust models are less over-confident. The full evaluation is included in our revised manuscript (Section C).
> - We evaluated the ROC curves (Figure 20) specifically for the improved downsampling and activation function and report the AUC values (Table 3) for the models in the appendix (Section E).
> - We fixed the Typo in the caption in Figure 7, according to the last point mentioned in the questions.

---

> > ### Comment · Reviewer_aQN4 · 2022-08-09
> > **Thanks for the response, my score stays the same**
> >
> > Dear authors,
> >
> > Thanks for the response and modifications to answer some of the concerns.
> >
> > Table 3 is a subset of AUCs that I wanted to see (like figure 4). Some of these can be included in the plots with close results. For example, the AUCs can be put in legends or titles of Figure 4 for Square attacks.
> >
> > Thanks for including non-robust results. Overall it seems that on unseen attacks the ROCs are close for robust vs non-robust models (Figure 4). Some of the CIFAR10-C results are mixed (saturate, contrast, brightness).
> >
> > I read the reviews from the other reviewers. I agree that other priori work hinted at this property of AT, but this paper's extensive experimentation methodology for establishing this fact is interesting and above borderline.
> >
> > Overall I don't change my score. The work is definitely above borderline, but the results on Squares and CIFAR10-C discourage me from a 7+ rating, given that robust models are marginally better than non-robust ones in detecting the attacks. The results on learnable activations and FLC are interesting, but still not a clear win.

---

### Official Review · Reviewer_92ac · 2022-07-11

**Rating:** 5
**Confidence:** 4
**Soundness:** 3 good
**Presentation:** 3 good
**Contribution:** 3 good

**Summary:**

This paper collected 71 robust models and their counterpart non-robust models, do inference on CIFAR-10, CIFAR-100, and ImageNet. They find that generally robust models have less confidence in both clean data and attacked data. They also find that downsampling strategies and activation functions influence much on prediction confidence.

**Questions:**

1. Obviously, high confidence in correct predictions and low confidence in incorrect predictions is good. But why low confidence in both correct and incorrect predictions is better than high confidence in both correct and incorrect predictions?

2. Are there any straight connections between downsampling strategies and prediction confidence?

**Limitations:**

An extensive report, but lacks novelty.

**Strengths And Weaknesses:**

Strength:
1. Writing is easy to follow.
2. Enough results to support their argument.

Weakness:
1. Just making statistics on off-the-shelf checkpoints and reporting the results. No novel designs of architectures or training strategies are proposed.
2. Models are simply classified as 'robust' and 'non-robust'. For different 'non-robust' models, there is less analysis of the adversarial training strategies.
3. The conclusion, that robust models are less overconfident, is not surprising. Robust models are trained on examples that maximize the loss function during adversarial training, so empirically they are less confident about their decisions.
4. Lack of detailed ablations of architecture influences.

======== Post-rebuttal Update =========
After a more thorough investigation, I do find that this idea is not covered by prior works. This work shows the lower-confidence phenomenon of robust models through large-scale contrast experiments, and also gives applications including predicting erroneous decisions and detecting adversarial samples. Based on these reconsiderations, I would like to raise my score to acceptance.

---

> ### Author Response · Authors · 2022-08-02
> **Response to Review**
>
> Thank you for your review of our paper. In the following, we will address the points mentioned under weaknesses and questions one-by-one.
>
> W1: [just using off-the-shelf checkpoints]  Actually, we do not only provide statistics on off-the-shelf checkpoints (see lines 154ff, Experimental Setup in our paper). We access checkpoints of different adversarially trained models from RobustBench, which we call “robust”. Thus, we understand a model to be robust, if it shows robust accuracy on RobustBench of more than 41.44 % on CIFAR10, 18.95% on CIFAR100 and 25.32 on ImageNet (listed on RobustBench) accuracy. All non-robust models are self-trained and have 0% robust accuracy! To facilitate the presented analysis, we train all 71 architectures with the respective training schemes to high clean validation accuracies as seen for example in Figure 1. Thus, our paper facilitates the first solid analysis of the behavior of robust versus non-robust models by providing direct comparison of 71 models. As stated in line 157, we will publish all trained models upon acceptance to facilitate future research.
>
> W2: [distinction between robust and non-robust models] We understand a model to be robust if it shows robust accuracy on RobustBench of more than 41.33 % accuracy on Cifar10. All non-robust models are self-trained and have 0% robust accuracy.  We specify this understanding of “robust models” versus “non-robust models” in the revision.
>
> W3: [observation is not surprising] While we agree with the reviewer’s intuition, we would like to point out that we are not aware of any prior theoretical proof or any other statistically significant empirical analysis which has actually shown this before. Our non-trivial contribution is to replace intuitions with a scientific analysis, providing a solid base for further works in model calibration and robustness.
>
> W4: [ablations of architecture influences] We have discussed this aspect already in the limitations section of our paper. We agree that our submission can only be understood as the starting point of analysis for exactly this reason. Yet, we want to point out that we show some more details on the behavior of FLC-pooling networks in the supplementary material in Figure 18.  Since training high-quality adversarially robust models is equally expensive as it is technically non-trivial, especially for novel architecture designs, we understand this first dataset of paired “robust” and “non-robust” models as an important first step, even though the variance in the underlying architecture design is still limited.
>
> Q1: The over-confidence, as defined by Naeini et al, [54] and in equation (3) of our paper, takes into account the confidence of models for incorrect predictions while disregarding the confidence in correct predictions. Low overall confidence can indeed reduce the model over-confidence, but this would lead to an increased calibration error. Please refer to Equations (1) to (3) to clarify.
>
> Q2: We are not aware of any detailed study that would provide such empirical evidence. Yet, following the argumentation of Grabinski et al., ECCV 2022, models with traditional pooling operations can suffer from aliasing and thus focus on high frequency information that is not reliable. This concept is in line with the implications of the widely discussed texture bias [2], as well as the robust model by Saikia et al [3]. We add this discussion to the revision of our paper.
>
> [2] Geirhos, R., Rubisch, P., Michaelis, C., Bethge, M., Wichmann, F. A., & Brendel, W. (2018). ImageNet-trained CNNs are biased towards texture; increasing shape bias improves accuracy and robustness. arXiv preprint arXiv:1811.12231.
> [3] Tonmoy Saikia, Cordelia Schmid, Thomas Brox, Improving robustness against common corruptions with frequency biased models, CVPR 2021.

---

> > ### Author Response · Authors · 2022-08-08
> > **Have your concerns been addressed?**
> >
> > The end of the discussion phase is approaching and we still did not receive any feedback on our response. In our rebuttal, we clarified that we dit **not only evaluate off-the-shelf models** but rather provide **71 newly trained models** to allow for an in depth and statistically significant analysis of paired robust and non-robust models.
> >
> > Based on your review, we understand that the low score was assigned due to an honest misunderstanding with respect to this point.  After our clarification, do you have any additional questions or concerns? If not, please consider updating our score accordingly.
> >
> > Please don't hesitate to let us know if you have any further questions or remarks!

---

> > > ### Comment · Reviewer_92ac · 2022-08-08
> > > **Response**
> > >
> > > Thanks for your response and revision. I do have a misunderstanding before about the models. But I still think there is not enough novelty for acceptance. Most results are known empirically, this work just verifies these in a more statistically scientific way, e.g., robust models have lower confidence, and confidence can be used in adversarial examples detection. So I would like to keep a borderline score.

---

> > > > ### Comment · Reviewer_pPVH · 2022-08-08
> > > > **Reviewer Response**
> > > >
> > > > I was not aware of the "less overconfident" results previously. What results are you thinking of when you write that the results in the paper are already known?

---

> > > > > ### Author Response · Authors · 2022-08-09
> > > > > **Thank you for asking this question**
> > > > >
> > > > > Dear reviewer,
> > > > > We are grateful that you raised this question. We are also not aware of any such study and we think that our paper makes an important contribution in this respect. Since we did not receive any answer from reviewer pPVH regarding this claim on missing novelty, we are depending on your final score. If you agree that our paper should be presented at NeurIPS 22, please consider to increase your rating to weak accept.

---

> > > > ### Author Response · Authors · 2022-08-09
> > > > **Inquiry on Reference**
> > > >
> > > > Dear reviewer 92ac,
> > > > thank you for the discussion! To further improve our submission, we would also be very interested in the mentioned reference! Which paper are you referring to, that analyzes the decrease in confidence of adversarially robust models? We would really like to include such work into our analysis - but are not aware of such prior study. If you don't have any references in mind, please consider raising your score to an accept score.
> > > > Thank you again for helping us to improve our submission!

---

> ### Author Response · Authors · 2022-08-05
> **Has our response addressed your concerns?**
>
> Dear Reviewer,
> Could we address your concerns in our rebuttal and revision? Please let us know in case we missed anything or in case you have any additional questions that we could address.
> Thanks!

---

### Official Review · Reviewer_itb7 · 2022-07-12

**Rating:** 3
**Confidence:** 4
**Soundness:** 2 fair
**Presentation:** 3 good
**Contribution:** 2 fair

**Summary:**

The paper studies the calibration abilities of robust models. The paper investigates 71 adversarially trained models and compares these with naturally trained counterparts. The paper observes that most non-robust models are over-confident but robust models are less confident so they are better calibrated. Additionally, the paper observes that specific layers, downsampling, and activation functions can lead to better calibration.

**Questions:**

The paper can be improved better if the authors can actually improve applications or existing methods based on the observation in this paper.


**Limitations:**

The limitation is discussed.


-- Post rebuttal

Thank the authors for their response. I am sorry for the reference, but this definitely does not affect my final score. Even though your claim is correct, I still think that the contribution of this paper is very marginal and not enough for acceptance. And after reading other reviews, it seems like other reviewers also agree on this point. Therefore, I kept my original score.

**Strengths And Weaknesses:**

The paper analyzes the calibration abilities thoroughly with many robust models and arrives at the conclusion. However, the observation is not very interesting as it is expected behavior. Additionally, technical contributions are limited. I do not believe the contribution of this paper is enough for top-tiered conferences including ICLR.

---

> ### Author Response · Authors · 2022-08-02
> **Response to Review**
>
> With all due respect, this review hardly meets the minimum standards one should expect from NeurIPS: it does not provide any helpful or constructive criticism. Instead, it reads like a template rejection phrase where the reviewer even forgot to insert the correct venue (we are not at ICLR (!)). Beyond this, the summary of our paper is also incorrect: While one would, from the spurious results of previous methods [0] and [1] mentioned by Reviewer pPVH and our reference [44], expect robust models to be better calibrated, our study shows that this is not necessarily the case (see for example Figure 3). Robust models are less overconfident, but in their training domain (clean samples, Figure 3, left), the calibration is better for non-robust models than the calibration of robust models. Only when adversarial examples or out-of-domain samples are considered (Figure 3, center and right, respectively), they actually show improved calibration. Yet, robust models are always less over-confident. We will state this more clearly in the revision.

---

> ### Author Response · Authors · 2022-08-09
> **final justification**
>
> Thank you for specifying your final justification. We understand that you want to follow the general trend of your fellow reviewers. In this context, we want to point out that the other reviewers have raised their score over the course of the discussion period and would ask you to do the same. Please also refer to their arguments in favor of our paper.

---

> > ### Comment · Reviewer_itb7 · 2022-08-09
> > **Response**
> >
> > Please, don't get the comment wrong. I do not just follow the trend. I explicitly stated that the novelty/contribution is not enough. I said I also found similar concerns on this point from other reviewers. And just curious, I see that only one reviewer raised the score, but others kept the original score. What makes you say "other reviewer**s** have raised their score"?

---

> > > ### Author Response · Authors · 2022-08-09
> > > **Response**
> > >
> > > Thank you for the response. Two reviewers changed their score: pPVH, 92ac. Therefore, it is fair to say that reviewer*s* changed their score*s*. Either way, we would like to spend the limited remaining time focusing on our paper and not rhetorics.
> > >
> > > You now have stated numerous times that you are having issues with "novelty/contribution" - Please provide references that back this statement and show a lack of novelty.

---

> > > ### Author Response · Authors · 2022-08-09
> > > **Regarding novelty**
> > >
> > > We would also like to point out our following comments regarding related work:
> > >
> > > https://openreview.net/forum?id=5K3uopkizS&noteId=nq_2pyo_YgA
> > >
> > > https://openreview.net/forum?id=5K3uopkizS&noteId=We6q89kvsWt
> > >
> > > https://openreview.net/forum?id=5K3uopkizS&noteId=p79-zaN84oN
> > >
> > > Please also pay attention that the other reviewers were unable to provide further references

---

### Official Review · Reviewer_pPVH · 2022-07-13

**Rating:** 5
**Confidence:** 4
**Soundness:** 2 fair
**Presentation:** 2 fair
**Contribution:** 2 fair

**Summary:**

In this work the authors investigate differences in calibration-related properties between standardly trained models and those with robustness interventions. Starting from the well known result that standardly trained models are over confident (in that they show high confidence even on incorrectly classified examples), the authors find that models with adversarial robustness interventions generally are less overconfident and better calibrated in terms of empirical calibration score, a heuristic for measuring calibration. They also find that while one can predict whether or not a model will correctly classify a given example about the same for both categories of models on natural test set examples, models trained with adv. interventions can much better predict whether or not an example that has been adversarially perturbed will be predicted correctly (both in the original threat model, and in an $\ell_0$ threat model). The authors claim these results as the basis for a newly discovered relationship between adversarial robustness and calibration.

**Questions:**

Questions are in strengths and weaknesses.

**Limitations:**

Yes

**Strengths And Weaknesses:**

The paper presents results that uncover interesting properties of robustly trained models. However, there are some concerns with treatment of prior work / novelty.

Prior work/novelty: there are two prior related papers connecting adversarial robustness and calibration:
- [0]: https://arxiv.org/abs/2006.16375 (NeurIPS 2021): finds that adversarially robust models have much better calibration properties according to the ECE heuristic (Section 3).
- [1]: https://arxiv.org/abs/1910.06259 (ICML 2021): Use calibration to perform adversarial training by reducing confidence on adv examples at train time.

[0] in particular looks like it contain results that could be highly related to the results presented here (i.e. seems to show a result very close to one of the core contributions in this paper); it would be good to clarify this situation.

---

> ### Author Response · Authors · 2022-08-02
> **Delimitation to previous work**
>
> Thank you for your valuable suggestions: We will add references to [0] and [1].
> Their initial experiments confirm the necessity of a solid empirical analysis of the relationship between model confidence, calibration and robustness.
> In particular, [0] do not investigate adversarially trained models but instead only look at vanilla networks and construct adversarial samples using the Carlini&Wagner attack. This is a highly limited perspective. They find that easily attackable data points are badly calibrated and that adversarial models have better calibration properties. In contrast, we analyze the robustness of models rather than individual data points on many paired samples.
> Importantly, the findings of our large scale study are more differentiated (see Figure 3). While robust models are always less over-confident, they are not always better calibrated with respect to clean (in-domain) data (Figure 3). Their calibration is much improved on the adversarial data they are trained on (this is less surprising). Their calibration is also improved with respect to out-of-domain data (Squares adversarial samples).
> [1] limit their analysis to one baseline model for which they report overconfident behavior of a robust model on SVHN. Further, they only report evidence for low resolution data. Their aim is to motivate the use of calibration to perform improved adversarial training.
>
> Instead, our submission facilitates a large-scale analysis of both, low resolution (CIFAR10/100) and high resolution models (ImageNet), on a substantial number of model pairs of robust and non-robust models. Given these remarks, we are the first to offer such a study and accompanying dataset and hope that it will be of broader use to the community.

---

> > ### Comment · Reviewer_pPVH · 2022-08-07
> > **Response**
> >
> > Thank you for the response. I can't see this discussion of related work in the paper---can you add a discussion?

---

> > > ### Author Response · Authors · 2022-08-07
> > > **more in-depth discussion**
> > >
> > > Thank you for your reply! Due to the page limitation in the revision, we have only added the references with a short discussion in the revised paper.  Specifically, we have added the mentioned reference [0] in line 78 (as our reference [58])  and briefly discussed [1] in line 136ff as our reference [70]. We fully agree that a more in-depth discussion such as the one provided in our initial reply will be helpful and the final paper template also offers some additional space. We would therefore gladly add the following discussion after line 133 of our paper:
> > >
> > > “Yet, only few but notable prior works such as [44,58] have investigated adversarial training with respect to model calibration. Without providing a systematic overview, [44] show that AT can help to smooth the predictive distributions of CNN models. Qin et al. [58] investigate adversarial data points generated using [5] with respect to non-robust models and find that easily attackable data points are badly calibrated while adversarial models have better calibration properties. In contrast, we analyze the robustness of models using paired model samples rather than investigating individual data points. Importantly, our proposed large-scale study allows a more differentiated view onto the relationship between adversarial training and model calibration, as discussed in Section 3. In particular, we find that adversarially trained models are not always better calibrated than vanilla models especially on clean data, while they are consistently less over-confident.”
> > >
> > > We hope that our response addresses all of your concerns. Thank you for your time and feedback on our submission! Please don't hesitate to let us know if you have any further suggestions!

---

> > > > ### Comment · Reviewer_pPVH · 2022-08-08
> > > > **Response**
> > > >
> > > > Thank you for the quick response and for making these changes, my concerns are addressed.

---

> > > > > ### Author Response · Authors · 2022-08-08
> > > > > **Score**
> > > > >
> > > > > Thank you for adjusting your score! Given that you indicated that we have addressed all your concerns: would you consider accepting the paper or could you elaborate what you feel is missing to further increase your score?

---

> ### Author Response · Authors · 2022-08-05
> **Has our response addressed your concerns?**
>
> Dear Reviewer,
> We did not receive any feedback on our response yet.
> Please let us know whether our rebuttal and revision have addressed your concerns and whether you have any additional questions or concerns we should address.
> Thanks!!

---

### Author Response · Authors · 2022-08-02
**Revised Manuscript**

We would like to thank all reviewers for their reviews and valuable suggestions. To focus your attention on the changes we have uploaded a  revised colour-coded (in orange) manuscript.

---

### Meta-Review · Area_Chair_rwgt · 2022-08-27

**Recommendation:** Accept
**Confidence:** Certain

**Metareview:**

This paper empirically demonstrates that adversarially trained models are better calibrated than naturally trained counterparts. The reviewer found this paper interesting, and initial concerns are mainly about 1) missing discussions of prior works, and 2) requiring more ablations.

The rebuttal well addresses most concerns (especially regarding the novelty w.r.t. prior works). As a result, three (out of four) reviewers unanimously agree to accept this submission. The reviewer itb7 is the only one against accepting this paper; nonetheless, the original review from itb7 is sort of vague and does not provide useful information for instructing authors for preparing a high-quality rebuttal accordingly. Also as the AC, I cannot see any significant concerns/drawbacks raised in the reviewer itb7's comments, therefore decide to ignore it.

In the final version, the authors should include all the clarifications and the additional empirical results provided in the rebuttal.


**Award:**

No

---

### Decision · Program_Chairs · 2022-09-14

Accept